

# Genomic diversity and evolution, diagnosis, prevention, and therapeutics of the pandemic COVID-19 disease

M. Nazmul Hoque[1,2], Abed Chaudhury[3], Md Abdul Mannan Akanda[4], M. Anwar Hossain[2,5] and Md Tofazzal Islam[6]

[1] Department of Gynecology, Obstetrics and Reproductive Health, Bangabandhu Sheikh Mujibur Rahman Agricultural University (BSMRAU), Gazipur, Bangladesh
[2] Department of Microbiology, University of Dhaka, Dhaka, Bangladesh
[3] Charles Sturt University, Orange, NSW, Australia
[4] Department of Plant Pathology, Bangabandhu Sheikh Mujibur Rahman Agricultural University (BSMRAU), Gazipur, Bangladesh
[5] Jashore University of Science and Technology, Jashore, Bangladesh
[6] Institute of Biotechnology and Genetic Engineering (IBGE), Bangabandhu Sheikh Mujibur Rahman Agricultural University (BSMRAU), Gazipur, Bangladesh

Corresponding author
Md Tofazzal Islam,
tofazzalislam@bsmrau.edu.bd,
tofazzalislam@yahoo.com

## ABSTRACT

The coronavirus disease 19 (COVID-19) is a highly transmittable and pathogenic viral infection caused by a novel evolutionarily divergent RNA virus, the severe acute respiratory syndrome coronavirus 2 (SARS-CoV-2). The virus first emerged in Wuhan, China in December 2019, and subsequently spreaded around the world. Genomic analyses revealed that this zoonotic virus may be evolved naturally but not a purposefully manipulated laboratory construct. However, currently available data are not sufficient to precisely conclude the origin of this fearsome virus. Comprehensive annotations of the whole-genomes revealed hundreds of nucleotides, and amino acids mutations, substitutions and/or deletions at different positions of the ever changing SARS-CoV-2 genome. The spike (S) glycoprotein of SARS-CoV-2 possesses a functional polybasic (furin) cleavage site at the S1-S2 boundary through the insertion of 12 nucleotides. It leads to the predicted acquisition of 3-*O*-linked glycan around the cleavage site. Although real-time RT-PCR methods targeting specific gene(s) have widely been used to diagnose the COVID-19 patients, however, recently developed more convenient, cheap, rapid, and specific diagnostic tools targeting antigens or CRISPR-Cas-mediated method or a newly developed plug and play method should be available for the resource-poor developing countries. A large number of candidate drugs, vaccines and therapies have shown great promise in early trials, however, these candidates of preventive or therapeutic agents have to pass a long path of trials before being released for the practical application against COVID-19. This review updates current knowledge on origin, genomic evolution, development of the diagnostic tools, and the preventive or therapeutic remedies of the COVID-19. We also discussed the future scopes for research, effective management, and surveillance of the newly emerged COVID-19 disease.

## INTRODUCTION

Emergence and reemergence of various pathogens pose global challenges for public health and human food security (*Islam et al., 2016*; *Gao, 2018*). The novel coronavirus SARS-CoV-2 has emerged as one of the deadliest viral human pathogens in last one hundred years after the Spanish Flu in 1918 (*Reid et al., 1999*). In late December 2019, the World Health Organization was notified of a cluster of cases of pneumonia disease of unknown etiology in Wuhan of Hubei Province of China. Soon afterwards, the researchers assumed that the culprit pathogen was a new coronavirus, which causes a severe acute respiratory syndrome in the infected patients. Based on phylogenomics and transmission electron microscopic analyses, *Zhou et al. (2020a)* first confirmed the pathogen as a novel coronavirus and named it as 2019-nCoV. Later, this new virus was renamed as SARS-CoV-2 and the disease caused by this virus was termed as COVID-19 by the Coronavirus Study Group of the International Committee on Taxonomy of Viruses (ICTV). The SARS-CoV-2 is the third devastating coronavirus (CoV) that infects human. Earlier, two similar zoonotic coronaviruses that emerged as epidemics to cause human infections were severe acute respiratory syndrome (SARS-CoV) in 2003 (*Zaki et al., 2012*; *Almofti et al., 2018*), and the Middle East respiratory syndrome (MERS-CoV) in 2012 (*Badawi et al., 2016*; *Pallesen et al., 2017*; *Ul Qamar et al., 2019*). Surprisingly, the COVID-19 disease rapidly almost spread to the whole world within a few months and poses a serious threat to human health globally. Considering the contagious behavior and fatality of the COVID-19, WHO declared it as a Public Health Emergency of International Concern (*WHO, 2020*). As of June 25, 2020, the COVID-19 has spread to 216 countries or territories, infecting at least 91,62,375 people of which around 4,73,087 died globally. The rapidly spreading person-to-person transmission of SARS-CoV-2 has been confirmed by detecting the virus in a wide range of samples including bronchoalveolar-lavage (*Zhu et al., 2020*; *Nishiura, Linton & Akhmetzhanov, 2020*), sputum (*Lin et al., 2020*), saliva (*To et al., 2020*), throat (*Bastola et al., 2020*) and nasopharyngeal swabs (*To et al., 2020*).

The SARS-CoV-2 belongs to the genus *Betacoronavirus* under the family *Coronaviridae*, is a positive-sense single-stranded RNA (+ssRNA) virus. The *Coronaviridae* is one of the largest viral families. Viruses under this family have potential ability to infect and subsequently cause diseases to a large number of mammals, birds, and humans (*Ahmed, Quadeer & McKay, 2020*; *Hemida & Abduallah, 2020*). The coronaviruses manifest a wide variety of clinical sign and symptoms, which include respiratory, nervous, enteric, and systemic health problems (*Hemida & Abduallah, 2020*; *Huang et al., 2020*). Within weeks of the first outbreak of COVID-19 disease in Wuhan, the complete genome sequence of this novel virus was published (*Zhou et al., 2020a*). Approximately, 30 kilobase sized genome of the novel SARS-CoV-2 encodes several smaller open reading frames (ORFs) (*Rota et al., 2003*; *Freundt et al., 2010*, *Cotten et al., 2013*). These ORFs encode for different proteins for example the replicase polyprotein, the spike (S) glycoprotein, envelope (E), membrane (M), nucleocapsid (N) proteins, accessory proteins, and other non-structural proteins (nsp) (*Ahmed, Quadeer & McKay, 2020*; *Islam et al., 2020*; *Phan, 2020*; *Walls et al., 2020*). The genome of SARS-CoV-2 coupled with regions of genomic instability

(*Abdelmageed et al., 2020*; *Rahman et al., 2020*), which encodes for multiple structural and non-structural proteins (*Ahmed, Quadeer & McKay, 2020*; *Rahman et al., 2020*) with many unique features. These features make these proteins prone to frequent coding changes, thus generating new strains in a short period of time (*Hemida & Abduallah, 2020*; *Islam et al., 2020*). Rapid mutational frequencies are associated with the poor proofreading efficiency of the viral RNA polymerase, and the likelihood of recombination between different members of this family (*Jackwood, Hall & Handel, 2012*; *Phan, 2020*). The relatively faster spread and varying levels of fatality of SARS-CoV-2 in different countries raises an intriguing question whether the evolution of this virus is driven by mutations. To address these question, several recent studies reported that substitution and/or deletion of nucleotides and amino acids (aa) at the entire genome of SARS-CoV-2 are the important mechanisms for virus evolution in nature (*Huang et al., 2020*; *Islam et al., 2020*; *Phan, 2020*; *Yin, 2020*). Due to the practice of open science, the research progress on SARS-CoV-2 is the fastest moving subject in the human history. In about six months, thousands of reports and data on genomics, origin, genome evolution, molecular diagnosis and vaccine and/or therapeutics of the SARS-CoV-2 have been published (*Clover, 2020*; *Geo-Vax, 2020*; *Islam et al., 2020*; *Phan, 2020*; *Rahman et al., 2020*; *Shereen et al., 2020*; *Shanmugaraj et al., 2020*; *Walls et al., 2020*; *Zhang et al., 2020a*).

Genomic analyses of the SARS-CoV-2 virus revealed that evolution of this virus is mainly driven by genetic drift and founder events (*Chiara et al., 2020*; *Huang et al., 2020*; *Islam et al., 2020*; *Yin, 2020*). Nevertheless, many researchers predicted a possible adaptation at the nucleotide, aa, and structural heterogeneity in the viral proteins, especially the spike (S) protein (*Armijos-Jaramillo et al., 2020*; *Islam et al., 2020*; *Sardar et al., 2020*). Recently, *Shen et al. (2020)* reported even an intra-host viral evolution during infection which might be related to its virulence, transmissibility, and/or evolution of virus response against the host immune system. To carry out its function, SARS-CoV-2 S protein binds to its receptor human angiotensin converting enzyme 2 (hACE2) through its receptor-binding domain (RBD), and is proteolytically activated by human proteases (*Shang et al., 2020a*). The efficient cell entry of the SARS-CoV-2 is mediated by the high hACE2 binding affinity of the RBD, furin preactivation of the spike, and hidden RBD in the spike while evading immune surveillance (*Shang et al., 2020a*). The virulence mechanisms of the SARS-CoV-2 are not fully understood (*Khan et al., 2020a*; *Zhou et al., 2020b*). It has been known that after cellular entry to the susceptible host, the SARS-CoV-2 manifests several clinical syndromes including pneumonia, fever, cough, shortness of breath, muscle pain (myalgias), fatigue, confusion, headache, sore throat, acute respiratory distress, and eventually multiorgan failure (*Jiang, Hillyer & Du, 2020*). Therefore, unravelling the cellular factors involved in entry of SARS-CoV-2 might give further insights into the transmission of the virus, and reveals the therapeutic targets (*Hoffmann et al., 2020a*; *Hemida & Abduallah, 2020*). However, the clinical sign and symptoms of SARS-CoV-2 in confirmed patients were highly variable. Therefore, the confirmatory diagnosis of COVID-19 is made with the aid of real-time reverse transcription–polymerase chain reaction (RT–PCR), computed tomography (CT)-scan, immune identification technology (Point-of-care Testing, POCT) of IgM/IgG, CRISPR-Cas or blood culture (*Ai et al., 2020*; *Corman et al., 2020*; *Hindson, 2020*; *Kellner*

*et al., 2019*; *Li et al., 2020a*; *Wang et al., 2020a*). Although RT-PCR is considered as gold standard, the development of new, low cost, convenient, rapid and specific diagnostic protocols are needed for monitoring, surveillance and management of this pandemic disease.

No effective therapeutic drugs or vaccines are yet to be discovered for the treatment of SARS-CoV-2 patients. Currently, some supportive cares are given to the patients such as oxygen therapy, antiviral combination with antibiotic, convalescent plasma therapy, antifungal treatment, and extra-corporeal membrane oxygenation (ECMO) (*Chen et al., 2020*; *Holshue et al., 2020*). Researchers across the globe are searching to find an antiviral drug useful in treating the infection of SARS-CoV-2. They evaluated several drugs or therapies namely, penciclovir, ribavirin, nitazoxanide, remdesivir (GS-5734), nafamostat, favipiravir (T-750) or Avigan, avermectins, dexamethasone, EIDD-2801, hydroxychloroquine, chloroquine, and convalescent plasma (CP) therapy against the infection of SARS-CoV-2 (*Duan et al., 2020*; *Liu et al., 2020*; *Martinez, 2020*; *Wang et al., 2020b*). The high mutation rate of the RBD leading to the faster evolution and high genomic disparity of this virus may help the new strains of this RNA virus to get away neutralization mechanism by RBD-targeting antibodies (*Rahman et al., 2020*). Therefore, non-RBD functional regions of the S glycoprotein could efficiently be selected for developing and devising effective therapeutic and prophylactic interventions against the infection by SARS-CoV-2. Several monoclonal antibodies (mAbs) with potent neutralizing activity targeting the N-terminal domain (NTD) of the S protein of SARS-CoV-2 has already been reported (*Shang et al., 2020b*; *Wang et al., 2019 Zhou etal, 2019*). In addition to S protein, two smaller proteins, E and M might also participate in the viral assembly of a coronavirus, and can mimic both cell-mediated and humoral immunity against SARS-CoV-2 (*Shi et al., 2006*; *Schoeman & Fielding, 2019*; *Shang et al., 2020b*). At least 90 vaccine candidates are now under trials for evaluating their efficacy and safety, and some of them are advanced to human trials (*Corey et al., 2020*).

Due to the practices of open science and open data sharing approaches, the literature generating through research on SARS-CoV-2 is simply explosive. The specific features of emerging pandemics, epidemiology, clinical characteristics, pathophysiology, diagnosis, treatment, ongoing clinical trials and prevention of the SARS-CoV-2 have been discussed in several reviews (*Guo et al., 2020*; *Tay et al., 2020*; *Tu et al., 2020*; *Valencia, 2020*; *Udugama et al., 2020*). However, no comprehensive review on the genomic diversity and evolution, diagnosis, prevention, and therapeutics of the SARS-CoV-2 has been published. Therefore, this report aims to review our current understanding on origin, genomic evolution, clinical and molecular diagnosis as well as prevention and control of the SARS-CoV-2 infection. Furthermore, this review also provides valuable information for further research and promotes responses of the relevant national and international authority to tackle this pandemic disease.

## REVIEW METHODOLOGY AND RATIONALE

From the very beginning of the first outbreak of SARS-CoV-2 in December, 2019 in Wuhan Province of China, thousands of reports and data on genomics, origin, genome

evolution, molecular diagnosis and vaccine and/or therapeutics of SARS-CoV-2 have been published. To prepare this review, we conducted a literature survey on the SARS-COV-2 in last six months. First, we focused the introduction section on the historical background of coronaviruses, genome composition and diversity, and progresses in the preventive measures against the SARS-CoV-2. We then searched the most up-to-date literature from PubMed central, Google Scholar, ResearchGate, bioRxiv, Preprints archives, China National Center for Bioinformation 2019 Novel Coronavirus Resource (2019nCoVR) and World Health Organization COVID-19 blog on the genome composition and diversity, genome-evolution and genome-wide mutations in SARS-CoV-2, diagnostic tools, proposed vaccine development, and therapeutics for COVID-19. We identified some important genome-wide mutations either at nucleotide or aa level that associated with the ever-changing phenomena of the virus irrespective of the geography and ethnicity. We also summarized the current acceptable theories on the emergence and evolution of SARS-CoV-2. Finally, we highlighted the progress to date in the control of SARS-CoV-2. Historically, the SARS-CoV-2 is the first pandemic affecting the entire globe with 216 countries or territories.

Though extensive research data on SARS-CoV-2 have been published by the researchers throughout the world, however, no comprehensive review on genomic diversity and evolution, diagnosis, prevention, and therapeutics of the SARS-CoV-2 has been published. This review will be useful for academicians, researchers and policymakers across the globe to better understand COVID-19, which will ultimately pave them a way for prevention and control of this pandemic disease.

## GENOMIC COMPOSITION OF THE SARS-COV-2

The positively-sensed single-stranded RNA SARS-CoV-2 virus (*Ahmed, Quadeer & McKay, 2020*) has a genome size of approximately 30 kb (range: 29.8 kb to 29.9 kb) (*Khailany, Safdar & Ozaslan, 2020*). It shares only about 80% sequence identity to the previously reported human coronaviruses (*Wu et al., 2020a*). The RNA molecule of the virus is surrounded by various proteins including S, M, E, and N (*Ahmed, Quadeer & McKay, 2020*). The genome of SARS-CoV-2 encodes for several smaller ORFs located in both in 5′-UTR and 3′-UTR regions of the genome (Figs. 1A– 1C) that are assumed to express eight new proteins termed as accessory proteins (*Rota et al., 2003*; *Freundt et al., 2010*). The 5′-UTR and 3′-UTR of the CoVs play vital roles in intra- and intermolecular interactions. They are functionally significant for RNA–RNA interactions, and for binding of viral and cellular proteins (*Yang & Leibowitz, 2015*). The first ORF at the 5′end is Pb1ab, which encodes for several non-structural proteins with the sizes of 29,844 bp (7,096 aa), 29,751 bp (7,073 aa) and 30,119 bp (7,078 aa) in SARS-CoV-2, SARS-CoV, and MERS-CoV, respectively. Differences at positions of 1,273 aa, 21,493 aa, and 1,270 aa in SARS-CoV-2, SARS-CoV, and MERS-CoV, respectively have been reported (*Mousavizadeh & Ghasemi, 2020*). Genetically, the SARS-CoV-2 is very less similar to SARS-CoV (about 79%) or MERS-CoV (about 50%) (*Mousavizadeh & Ghasemi, 2020*). The genomic position of the E, M, and N proteins among betacoronaviruses are different as depicted in Fig. 1. The

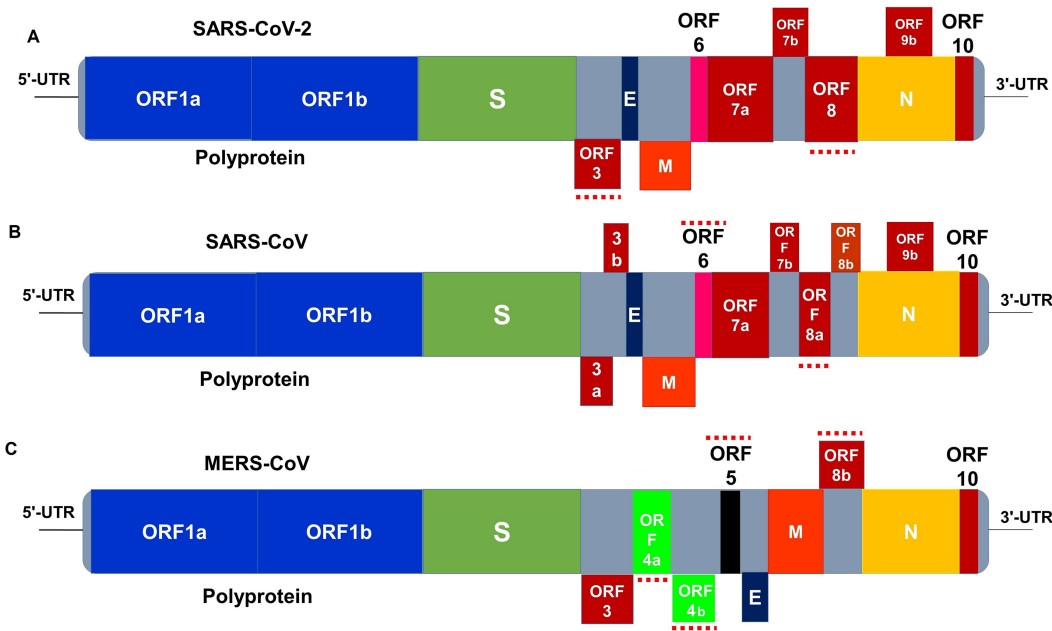

**Figure 1  Genome organization of (A) SARS-CoV-2, (B) SARS-CoV and (C) MERS-CoV.** The genome of these three viruses comprises the 5′-untranslated region (5′-UTR), polyprotein with open reading frame (orf) 1a/b (blue box) representing non-structural proteins (nsp) for replication, structural proteins including S glycoprotein (dark green box), envelop (E) (dark blue box), membrane (M) (orange box), and nucleocapsid (N) (yellow box) proteins, accessory proteins such as orf 3a/b (red boxes), 5 (black box), 6 (pink box), 7a/b, 8a/b, 9b and 10 (red boxes), and the 3′-untranslated region (3′-UTR). The doted red lines (both in above and under) are the protein which show key differences among SARS-CoV-2, SARS-CoV and MERS-CoV. The nsps and orfs lengths are not drawn in scale (adapted from *Islam et al. 2020*; *Shereen et al. 2020*).

accessory proteins are labelled as ORFs 1a and 1b (polyprotein), 3a, 3b, 6, 7a, 7b, 8a, 8b, 9b and 10 (*Figs. 1A–1C*). The size of these ORFs range from 39 to 274 aa (*Marra et al., 2003*; *Freundt et al., 2010*). These ORFs also encode for the replicase polyprotein, structural proteins, and other non-structural proteins (nsp) (*Ahmed, Quadeer & McKay, 2020*; *Walls et al., 2020*; *Phan, 2020*). The *orf1ab* is the largest gene in SARS-CoV-2, which encodes the polyprotein (pp1ab) and 15 nsps. The *orf1a* gene encodes for pp1a protein which also contains 10 nsps (*Shereen et al., 2020*). Noticeable differences between SARS-CoV and SARS-CoV-2 genomes such as absence of 8a protein and fluctuation in the number of aa in 8b and 3c protein in SARS-CoV-2 have been reported in several studies (*Shereen et al., 2020*; *Wu et al., 2020a*).

The CoVs use their S glycoprotein, a main target for antibody neutralization, to bind their receptor, mediate membrane fusion and entry into the host cell. Each monomer of homotrimeric S protein is about 180 kDa in size, which contains S1 and S2 subunits for mediating attachment and membrane fusion, respectively. The N- and C- terminal portions of S1 comprises two major domains S1 fold as two independent domains, the RBD and N-terminal domain (NTD) (*Song et al., 2018*; *Ou et al., 2020*; *Rahman et al., 2020*). While RBD of mouse hepatitis virus (MHV) is located at the NTD (*Kubo, Yamada & Taguchi,*

*1994*), most of other CoVs, including SARS-CoV and MERS-CoV use C-domain to bind their receptors (*Li et al., 2005*; *Lu et al., 2013*; *Ou et al., 2020*). During the pathogenesis, the trimeric S protein is cleaved into S1 and S2 subunits, and the RBD of the S1 subunit directly binds to the peptidase domain (PD) of ACE2 while the S2 carried out membrane fusion activity (*Yan et al., 2020*). Structural and biochemical studies revealed that the SARS-CoV-2 has an RBD which binds with high affinity to ACE2 from humans, ferrets, cats and other species with high receptor homology (*Andersen et al., 2020*; *Wan et al., 2020*; *Walls et al., 2020*; *Wrapp et al., 2020*; *Zhou et al., 2020a*). Therefore, the RBD of SARS-CoV-2 is a particularly snug fit, and 10–20 times more likely to bind ACE2 than SARS-CoV (*Wrapp et al., 2020*). Due to these novel genomic features (i) SARS-CoV-2 arises to be optimized for binding to the human ACE2 receptor; and (ii) the S protein of SARS-CoV-2 possesses a functional polybasic (furin) cleavage site at the S1–S2 boundary by way of the insertion of 12 nucleotides (*Walls et al., 2020*), which additionally led to the assumed acquisition of 3-O-linked glycans around the site. Moreover, this polybasic cleavage site "RRAR" is unique in SARS-CoV-2, rendering by its unique insert of "PRRA", and might have evolved from other human betacoronaviruses, including HKU1 (lineage A), and MERS-CoV (*Andersen et al., 2020*). Proteolytic cleavage sites of the S protein can determine whether the virus is evolved from a cross species, e.g., from bats to humans (*Andersen et al., 2020*). However, the functional furin cleavage site is absent in related 'lineage B' betacoronaviruses like the bat coronavirus strain, RaTG13 (*Andersen et al., 2020*; *Coutard et al., 2020*). Functional polybasic cleavage at the S1/S2 site is essential for spike-driven viral entry into lung cells (*Hoffmann, Kleine-Weber & Pöhlmann, 2020b*). *Lau et al. (2020)* suggested that the unique cleavage PRRA motif under strong selective pressure could promote SARS-CoV-2 infection in humans. Moreover, the S protein of SARS-CoV-2 encodes 22 N-linked glycan sequons per protomer, which play a role in protein folding and immune evasion. The SARS-CoV-2 S glycans differ from typical host glycan processing, and therefore, might have implications in viral pathobiology and vaccine design (*Watanabe et al., 2020*).

## GENOME EVOLUTION OF THE SARS-COV-2

Phylogenetic comparison of coronavirus sequences from the patients of different geographical regions, and climatic conditions supports the natural origin of SARS-CoV-2 (*Adachi et al., 2020*; *Andersen et al., 2020*; *Lu et al., 2020*; *Shereen et al., 2020*; *Zhou et al., 2020a*). The complete genomes of the novel SARS-CoV-2 sequenced from different patients share more than 99.9% sequence identity (*Tang et al., 2020*) suggesting a very recent host shift of this virus to humans (*Lu et al., 2020*; *Tang et al., 2020*; *Zhou et al., 2020b*). The genomic analysis revealed that the whole genome of SARS-CoV-2 shares 98.0%, 79.0% and 50.0% identity to the genomes of bat SARS-related coronavirus, Bat-SARSr-CoV-RaTG13, SARS-CoV and MERS-CoV, respectively (*Andersen et al., 2020*; *Coutard et al., 2020*; *Lu et al., 2020*; *Ou et al., 2020*; *Tang et al., 2020*; *Xiao et al., 2020*; *Zhou et al., 2020a*). SARS-CoV-2 related coronaviruses have also been identified in Malayan pangolins (*Lam et al., 2020*). Pangolin-CoV is 91.02% and 90.55% identical to SARS-CoV-2 and BatCoV RaTG13, respectively (*Lam et al., 2020*; *Tang et al., 2020*; *Xiao et al., 2020*). The trimeric S protein of

SARS-CoV-2 and SARS-CoV are phylogenetically closely related showing about 77% aa sequence identity (*Rahman et al., 2020*; *Yuan et al., 2020*; *Zhou et al., 2020b*). Furthermore, the RBD sequence of SARS-CoV-2 is very close (99%) to that of a pangolin coronavirus (*Lam et al., 2020*; *Tang et al., 2020*). These findings therefore suggest that SARS-CoV-2 is the result of the recombination of two viruses, and contains no trace of any human-mediated genetic manipulation. Thousands of complete genome sequences of the SARS-CoV-2 have already been deposited to the global database repositories including National Center for Biotechnology Information (NCBI), GSAID (global initiative on sharing all influenza data), and China National Center for Bioinformation 2019 Novel Coronavirus Resource (2019nCoVR) from the entire world. Phylogenetic analysis revealed that most of the SARS-CoV-2 strains from India correspond to those strains isolated from China. The Brazilian (EPI_ISL_417034/Brazil/2020), Australian (EPI_ISL_416412/Australia/2020), and Canadian (EPI_ISL_418827/Canada/2020) SARS-CoV-2 strains also showed neighboring relationship to the Indian and Chinese strains (Fig. 2). Moreover, one Nepalese SARS-CoV-2 strain (EPI_ISL_410301/Nepal/2020) showed close phylogenetic association with a Spanish strain (EPI_ISL_418244/Spain/2020). We also found a close similarity between a South American SARS-CoV-2 strain (EPI_ISL_418262/Columbia/2020) and a North American strain (EPI_ISL_42078/USA/2020) (Fig. 2). The genomic analyses of these sequences showed that some are genetically identical to each other, while others carry some distinctive mutations (*Islam et al., 2020*; *Phan, 2020*). Analyzing 200 whole genome sequences of the SARS-CoV-2 retrieved from the GISAID (https://www.gisaid.org/), we found that the evolution of this virus is not country or territory specific rather patient or ethnic group specific (Fig. 2). The ongoing pandemic outbreak of the SARS-CoV-2 indicates its alarmingly rapid transmission across the globe. Determining the origin and evolution of the SARS-CoV-2 is important for the surveillance, development of effective interventions for controlling the epidemic, and prevention of the SARS-CoV-2. Analyses of the novel SARS-CoV-2 genome and functional structures are needed to better understand its molecular cross-talks with human host (*Rahman et al., 2020*; *Zhang et al., 2020a*). Regular publication of pathogenic SARS-CoV-2 isolates in open science and open data sharing model, reexamination of their origin and diversification patterns are becoming clear. From the initial study on Wuhan COVID-19 outbreak to its rapid spread to more than 216 countries or territories in the world, researchers suggested that this novel virus is likely to have moved to human from bats via an intermediate host pangolin through host jump (*Zhou et al., 2020a*; *Wu et al., 2020b*; *Li et al., 2018*; *Sun et al., 2020*). Despite having 77.38% and 31.93% sequence uniqueness among the S proteins of the SARS-CoV and MERS-CoV, respectively (*Rahman et al., 2020*), the SARS-CoV-2 exhibited rich genetic diversity and frequent recombination events that might have increased the potential for its cross-species transmission (*Islam et al., 2020*; *Song et al., 2005*; *Sun et al., 2020*; *Zhou et al., 2020b*). The aa sequence of the RBD segment of the SARS-CoV-2 genome is 74% and 90.1% homologous to that of SARS-CoV and RaTG13, respectively (*Ou et al., 2020*). The genome-wide phylogenetic analysis indicated that SARS-CoV-2 is closest to RaTG13, followed by GD Pangolin SARSr-CoV, GX Pangolin SARSr-CoVs, ZC45 and ZXC21, human SARS-CoV, and BM48-31 (*Tang et al., 2020*).
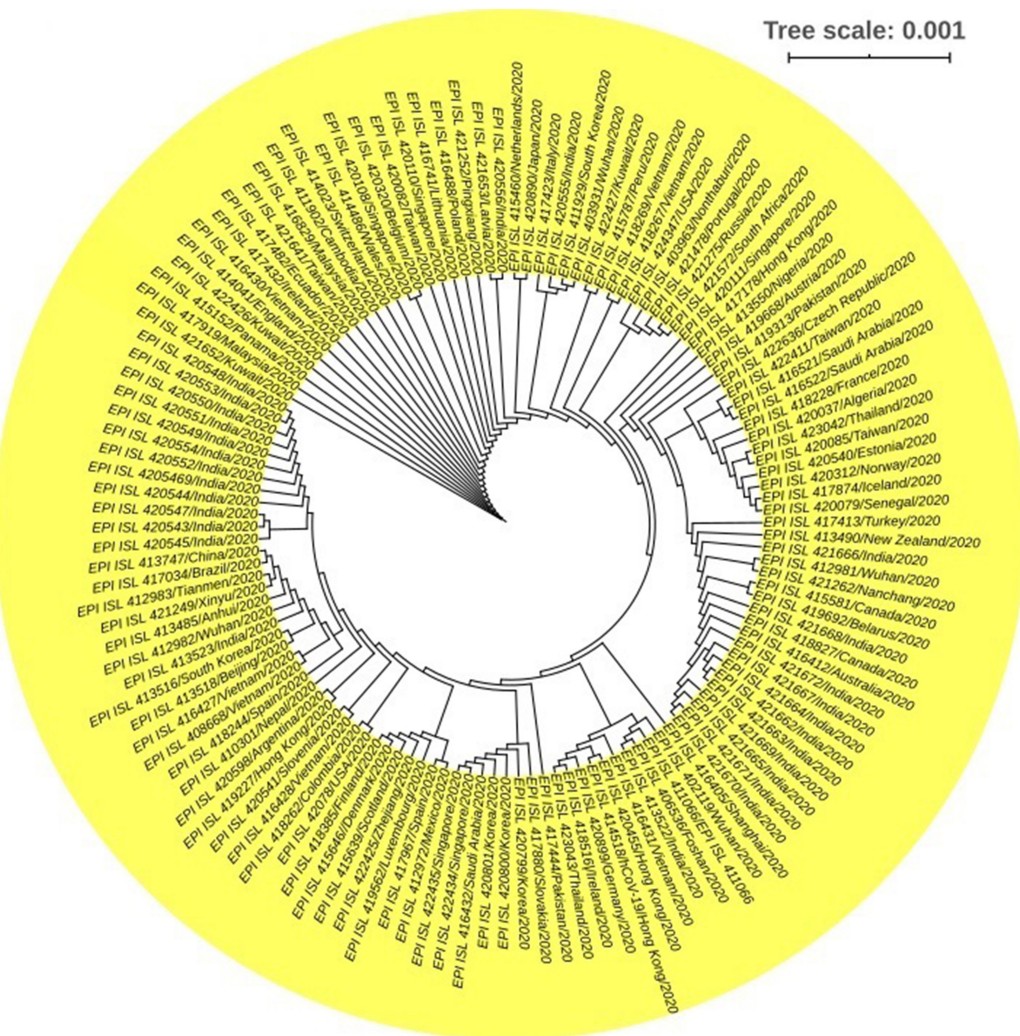

**Figure 2** **Phylogenetic tree of SARS-CoV-2.** 200 complete genome sequences of SARS-CoV-2 retrieved from global initiative on sharing all influenza data (GISAID) (https://www.gisaid.org/) from different countries were used to build this tree. The sequences were aligned using MAFFT online server (*Katoh, Rozewicki & Yamada, 2019*), and a maximum likelihood tree was built with iTOL (interactive Tree Of Life). Each node represents a single strain which is found to be patient and/or sample specific, and not clustered according to geographical locations. Tree scale 0.01, represents days before the time of lastly sampled genomes by scale*365.

Phylogenetic analysis of the recently released genomes of SARS-CoV-2 to the GISAID (https://www.gisaid.org/) revealed that the bats' CoV and the human SARS-CoV-2 shares a common ancestor (*Andersen et al., 2020*; *Zhang et al., 2020b*). These considerations indeed led the researchers and virologists around the globe to phylogenetically classify the SARS-CoV-2 as a SARS-like virus (*Zhang et al., 2020b*). In another study, *Sun et al. (2020)* reported that the SARS-CoV-2 shares a most recent common ancestor with BetaCoV/RaTG13/2013 (EPI_ISL_402131) due to their clustering in the same position. Conversely, *Lam et al. (2020)* demonstrated that the multiple putative lineages of pangolin

CoV sequences shared 85.5% to 92.4% similarity to SARS-CoV-2. Based on these similarities, they assumed that pangolins served as a potential intermediate host (*Lam et al., 2020*; *Sun et al., 2020*). In a phylogenetic network analysis of 160 complete human SARS-CoV-2 genomes, *Forster et al. (2020)* reported three central variants (A, B, and C) distinguished by aa changes, which we have named A, B, and C, with A being the ancestral type according to the bat outgroup coronavirus. The A and C types belonged to the Europeans and Americans while the B type is the most common type in East Asia (*Forster et al., 2020*). *Boni et al. (2020)* reported that the ancestors of SARS-CoV-2 separated from the bat version, which subsequently lost the effective RBD that was present in its ancestors (and remains in SARS-CoV-2). Two circumstances can plausibly explain the origin of SARS-CoV-2: (i) natural selection in humans following zoonotic transfer; and (ii) natural selection in an animal host before zoonotic transfer. However, currently available data are not sufficient enough to precisely conclude whether the virus was directly transmitted from bats to humans or indirectly through an intermediate host, pangolin. Inevitably, we need more sequence data to confirm the specific genetic identity and the origin of the SARS-CoV-2, which can be achieved by improved collection and monitoring of human samples across the globe, bat, pangolin and other wild animal samples as well.

## GENOME-WIDE MUTATIONS INFER THE EVOLUTION OF SARS-COV-2 VARIANTS

Mutations in the viral genomes are considered as the building blocks of their evolution that remain as the key factor to the novelty in evolution (*Baer, 2008*; *Duffy, 2018*). RNA viruses like SARS-CoV-2 are perhaps the most intriguing biological entities to adapt to new environments, possesses traits considered beneficial for them and higher mutation rates, which are correlated with enhanced virulence and evolvability (*Carrasco-Hernandez et al., 2017*; *Duffy, 2018*; *Islam et al., 2020*). The ongoing rapid human to human transmission, and global spread of SARS-CoV-2 have raised some exciting questions, such as whether the evolution and host adaptation of this virus are driven by mutations. The inherently high mutation rates of SARS-CoV-2 has already produced many mutant clouds of descendants that complicates the conception of its genotyping. According to the nucleotide C28144T variation, the SARS-CoV-2 can be divided into group A (117 strains) and group B (256 strains) (*Zhang et al., 2020b*). Based on the variation of 11 nucleotide (nt) sites, *Zhang et al. (2020b)* speculated that the Washington strain is more like an ancestor type, and the Wuhan strain is the offspring of the group A virus strain. Nonetheless, several reports predicted the possible effects of genomics mutations, aa variations, and structural heterogeneity (Table 1) in the entire genomes of different strains of SARS-CoV-2 (*Andersen et al., 2020*; *Huang et al., 2020*; *Islam et al., 2020*; *Lu et al., 2020*; *Phan, 2020*; *Yin, 2020*; *Walls et al., 2020*).

Recently, *Islam et al. (2020)* reported 1,516 nucleotide (nt) mutations at different positions throughout the SARS-CoV-2 genome, and twelve deletion-sites in polyprotein ($n = 9$), ORF10 ($n = 1$) and 3′-UTR ($n = 2$) (Table 1). Through a systemic gene-level mutational analysis, 744 amino acid (aa) substitutions (*Islam et al., 2020*) in different ORFs, 16 aa substitutions at twelve positions (*Yuan et al., 2020*), 935 aa replacements in the

**Table 1  Genome-wide nucleotide mutations and amino-acid mutations and substitutions in SARS-CoV-2 strains.** The number in the parentheses indicated the missense mutations.

| Genome-site/position | No. of amino-acid replacements | No. of nucleotide mutations | References |
|---|---|---|---|
| Polyprotein (nsp) | 412 | 661 | *Islam et al. (2020)* |
| | 757 | | *Yin (2020)* |
| Leader sequence | 178 | | *Yin (2020)* |
| Spike (S) glycoprotein | 120 | 183 | *Islam et al. (2020)* |
| | 14 (8) | | *Phan (2020)* |
| | 183 | | *Yin (2020)* |
| | 7 | 11 | *Wang et al. (2020a)* |
| | 13 | | *Huang et al. (2020)* |
| | 18 | | *Lu et al. (2020)* |
| | 6 | | *Andersen et al. (2020)* |
| Membrane (M) protein | 15 | 34 | *Islam et al. (2020)* |
| | 2 (1) | | *Phan (2020)* |
| | 33 | | *Yin (2020)* |
| | | 5 | *Wang et al. (2020a)* |
| | 2 | | *Huang et al. (2020)* |
| Envelop (E) protein | 11 | 27 | *Islam et al. (2020)* |
| | 2 | | *Huang et al. (2020)* |
| Nucleocapsid (N) protein | 82 | 148 | *Islam et al. (2020)* |
| | 7 (4) | | *Phan (2020)* |
| | 6 | 17 | *Wang et al. (2020a)* |
| | 5 | | *Huang et al. (2020)* |
| | 222 | | *Yin (2020)* |
| Open-reading frames (ORFs) | | | |
| ORF1a | 44 | | *Huang et al. (2020)* |
| ORF1ab | 48 (29) | | *Phan (2020)* |
| ORF1ab | 8 | | *Huang et al. (2020)* |
| ORF1ab | 6 | 43 | *Wang et al. (2020a)* |
| ORF3a | 48 | 92 | *Islam et al. (2020)* |
| | 49 | | *Yin (2020)* |
| | 7 | | *Huang et al. (2020)* |
| | 6 | 6 | *Wang et al. (2020a)* |
| ORF6 | 5 | 8 | *Islam et al. (2020)* |
| ORF7a | 22 | 46 | *Islam et al. (2020)* |
| | 2 | | *Huang et al. (2020)* |
| ORF7b | 4 | 8 | *Islam et al. (2020)* |
| ORF8 | 16 | 33 | *Islam et al. (2020)* |
| | 8 | | *Huang et al. (2020)* |
| | 34 | 34 | *Wang et al. (2020a)* |

**Table 1** (*continued*)

| Genome-site/position | No. of amino-acid replacements | No. of nucleotide mutations | References |
|---|---|---|---|
| ORF10 | 10 | 17 | *Islam et al. (2020)* |
|  | 1 |  | *Huang et al. (2020)* |
| 5′-UTR |  | 105 | *Islam et al. (2020)* |
|  |  | 8 | *Phan (2020)* |
| 3′-UTR |  | 158 | *Islam et al. (2020)* |
|  | 3 |  | *Phan (2020)* |
| 3′-to-5′exonuclease | 62 |  | *Yin (2020)* |
| Spacer region |  | 6 | *Islam et al. (2020)* |
|  | 6 |  | *Phan (2020)* |

**Notes.**

Here nsp, non-structural proteins; ORF, open-reading frames; UTR, untranslated region.

polyprotein, and 183, 33 and 222 aa substitutions in the S, M and N proteins, respectively (*Yin, 2020*) have been reported (Table 1), which could have made the viral proteins heterogeneous. In a recent study, van Dorp and co-authors reported 198 mutations that appear to have independently occurred more than once, which may hold clues to how the virus is adapting (*van Dorp et al., 2020*). *Islam et al. (2020)* reported 12 aa substitutions in the RBD at 331 to 524 residues of S1 subunit in different SARS-COV-2 strains of Wales, USA, Shenzhen, Hong Kong, Shanghai, Guangdong, Finland, and France. Similarly, (*Sardar et al., 2020*) identified a unique mutation in the S glycoprotein (A930V) in the Indian SARS-CoV-2 strain, which was absent in other related SARS-CoV-2 strains from different geographical regions.

Six corresponding RBD aa (residue positions: Y442, L472, N479, D480, T487 and Y4911 in SARS-CoV, and L455, F486, Q493, S494, N501 and Y505 in SARS-CoV-2) have been reported to be critical for binding to ACE2 receptors, and determining the host range (*Andersen et al., 2020*; *Islam et al., 2020*). On the other hand, *Andersen et al. (2020)* reported that five of these six residues differ between SARS-CoV-2 and SARS-CoV. The RBD region (aa position: 338-530) of the SARS-CoV-2 genome individually faced aa mutations at 72 different positions in 394 strains, and the S1 and S2 subunits of the spike protein undergo 331 and 274 number of positional mutations, respectively (*Wrapp et al., 2020*). Mutations, insertions and deletions can occur near the S1–S2 junction of coronaviruses, which shows that the polybasic cleavage site can arise by a natural evolutionary process (*Andersen et al., 2020*). The aa substitutions related to asparagine in the RBD, and/or in S1/2 subdomains adjacent to the glycosylated sites may affect the glycosylation shield, folding of S protein, host-pathogen interactions, viral entry and finally immune modulation, thus antibody recognition and viral pathogenicity (*Ou et al., 2020*; *Watanabe et al., 2020*). Three mutation types circulating in Wuhan, Shenzhen, Hong Kong, and France, displayed enhanced structural stability along with higher human ACE2 receptor affinity of the S protein, indicating these mutants may have acquired increased infectivity to humans (*Ou et al., 2020*; *Wang et al., 2020c*). It is likely that a high mutation rate in S protein, coupled with strong natural selection, has shaped the identical functional aa residues between SARS-CoV-2 and GD Pangolin-CoV, as proposed previously (*Lam et al., 2020*; *Tang et*

*al., 2020*). In addition to site-specific mutations in the spike protein, several deletions in the ranged nucleotides were also reported in the polyprotein, ORF10 and 3′-UTR of the genome of SARS-CoV-2 strains reported from Japan, USA, England, Canada, Netherlands, Wuhan and Australia (*Islam et al., 2020*). The single N501T mutation in SARS-CoV-2's S protein may have significantly enhanced its binding affinity for ACE2 (*Shereen et al., 2020*). Furthermore, deletion of 5 aa (675-679 aa: QTQTN) at the upstream of the polybasic cleavage site of S1-S2, and 21 nt at 23596–23617 positions in the polybasic cleavage site in clinical samples and cell-isolated virus strain likely benefit the SARS-CoV-2 replication or infection *in vitro*, and also strong purification selection *in vivo* (*Liu et al., 2020*). These mutations, deletions and/or substitutions in the polyprotein, S, M and E proteins of the SARS-CoV-2 genome can potentially influence the tertiary structures and functions of the associated proteins, and ultimately affect the viral adaptation to human, host-virus interactions, attenuation, pathogenicity, and immune-modulations (*Islam et al., 2020*; *Phan, 2020*; *Xu et al., 2020*; *Qu et al., 2020*; *Zhou et al., 2020b*).

The emerging rapid community transmission, and global spread of COVID-19 have raised intriguing questions whether the evolution and adaptation of the SARS-CoV-2 in diverse geographic and climatic conditions driven by aa mutations, deletions and/or replacements (*Bal et al., 2020*; *Islam et al., 2020*; *Pachetti et al., 2020*). Hitherto, the exact role of geo-climatic condition on global pandemics of SARS-CoV-2 is largely unknown. Nevertheless, it would be worth keeping in mind that this novel disease originated from the wildlife before they spread to humans (*Harvey, 2020*). The ability of the different strains of SARS-CoV-2 strains for swift adaptations to the diverge environments could be linked to their geographical distributions. Conversely, phylogenomic analysis of three super-clades (S, V, and G) isolated from the outbreaks of distinct geographic locations (China, USA and Europe) could not clearly reflect the hypothetical ongoing adaptation of SARS-CoV-2, which alternately refer to mere genetic drift and founder effects due to rapid spreading of the virus (*Chiara et al., 2020*). Though not yet studied well, evidences suggested that the transmission of SARS-CoV-2 infections and per day mortality rate from this infection is positively associated with weather conditions, and diurnal temperature range (DTR) (*Brassey et al., 2020*; *Su et al., 2020*).

## DIAGNOSTIC TOOLS FOR THE COVID-19

The clinical symptoms expressed by SARS-CoV-2 patients are non-specific, and thus, cannot be used for an accurate diagnosis. Only molecular techniques are able to specifically detect specific pathogen in a convenient way. A rapid, specific and convenient diagnostic protocol might play a vital role in the containment of the SARS-CoV-2, helping the rapid implementation of management of the disease that limit the spread through case identification, isolation, and contact tracing (*Drew et al., 2020*). The complete genome sequence data of the virus was publicly available within weeks of the first outbreak in Wuhan. It helped researcher to target specific genes for the development of nucleic acid test within three weeks. The on-going outbreaks of SARS-CoV-2 could also be diagnosed more accurately using metagenomics approaches in a wider range clinical samples like other infectious diseases (*Hoque et al., 2019*; *Lam et al., 2020*).

The first real-time RT-PCR assays targeting 3 genes, nucleocapsid (*N*), envelop (*E*) and RNA-dependent RNA polymerase (*RdRp*) were developed and published on 23 January 2020 by *Corman et al. (2020)*. The *RdRp* gene of the SARS-CoV-2 genome is highly similar to that gene of bat coronavirus RaTG13 (*Zhou et al., 2020a*). Later consistent detection of SARS-CoV-2 in saliva was published by *To et al. (2020)*. Several groups and countries developed many diagnostic protocols targeting or using nucleic acid tests or protein/antibody, loop-mediated amplified technique, imaging techniques (CT-scan) or CRISPR-Cas mediated technology (Table 2) (*Broughton et al., 2020*; *Zhang, Abudayyeh & Gootenberg, 2020*). Recently, more and more user-friendly molecular tests are on the horizon for SARS-CoV-2 RNA screening, as for example using Heating Unextracted Diagnostic Samples Obliterate Nuclease, and cards to run Clustered Regularly Interspaced Short Palindromic Repeats (CRISPR) methods (*Broughton et al., 2020*; *Li et al., 2019*).

To diagnose the SARS-CoV-2, the real-time PCR (RT-PCR) method has been developed by several groups targeting different genes. For example, *Chan et al. (2020)* developed three methods of RT-PCR, and of these assays, the COVID-19-*RdRp*/Hel (RNA-dependent RNA polymerase (*RdRp*)/helicase) assay had the lowest limit of detection *in vitro* (1.8 TCID50/ml with genomic RNA and 11.2 RNA copies/reaction with *in vitro* RNA transcripts). This method was validated in testing 273 suspected patients where 15 patients were confirmed as SARS-CoV-2 positive. This method targeted the *RdRp*/Hel, *S*, and *N* genes of SARS-CoV-2 with that of the reported RdRp-P2 assay which is used in more than 30 European laboratories. Huge improvements have been achieved in the RT-PCR methods since it first development. However, there are some drawbacks of the RT-PCR, as for example kits can give some false-negative results, dependency on swab sampling and extraction method, and required highly skilled personnel, sophisticated facilities and equipment (*Nuccetelli et al., 2020*). The non-invasive radiographic technique, CT-scan, is more sensitive than RT-PCR, and has been widely used worldwide for the detection of SARS-CoV-2 (*Nuccetelli et al., 2020*). In fact, the chest radiograph assessment of the SARS-CoV-2 patients resembled many features of community-acquired pneumonia (CAP) that are similar to other organisms including SARS-CoV and avian influenza A H5N1 (*Cheng et al., 2004*). Through analysis of the data of 1,014 patients in China, CT scan was found to be sensitive than RT-PCR for diagnosis of SARS-CoV-2 (*Ai et al., 2020*). The chest CT imaging showed higher positive rates (88%, 888/1014) in diagnosing the COVID-19 suspected patients compared to the confirmatory rates (59%, 601/1014) of RT-PCR assays. The sensitivity of chest CT imaging for COVID-19 was 97%, where RT-PCR was used as a standard reference (*Ai et al., 2020*).

Developing plug-and-play diagnostics to manage the SARS-CoV-2 outbreak would also be useful in preventing future epidemics. A recently developed Abbott ID Now™ COVID-19 test has been found to be very convenient, and can detect SARS-CoV-2 in 5 min only. Similarly, several serological assays have been developed since the beginning of COVID-19 pandemic, including point-of-care test (POCT)-fluorescence assays, enzyme-linked immunosorbent assays (ELISA), rapid antibody immunochromatographic tests, and chemiluminescence immunoassays (CLIAs) (*Nuccetelli et al., 2020*). Serological tests are cheaper than molecular tests, require a shorter analytical time, and productivity can be much greater than molecular tests. However, these tests to detect antibodies against viral

Hoque et al. (2020), *PeerJ*, DOI 10.7717/peerj.9689

**Table 2  Diagnostic protocols developed for SARS-CoV-2.**

| Type of clinical sample | Method/platform (technology) | Target gene/ Biomarker | Who developed | References |
|---|---|---|---|---|
| Upper and lower respiratory specimens[a] | Real-Time RTPCR | *N* gene | U.S. CDC | *Anonymous (2020a)* |
| Upper and lower respiratory specimens[a] | Real-Time RTPCR | *ORF1ab* and *N* gene | China, CDC | *Anonymous (2020b)* |
| Respiratory specimens | Real-Time RTPCR | *RdRp*, *E* and *N* genes | Multicountri es: Germany, The Netherlands, China, France and UK | *Corman et al. (2020)* |
| Respiratory specimens | Real-Time RTPCR | *RdRp*/Hel, *S* and *N* genes | Hong Kong, China | *Chan et al. (2020)* |
| Saliva | Real-Time RTPCR | *S* gene | Hong Kong | *To et al. (2020)* |
| Human clinical specimen | Real-Time RTPCR | *ORF1b-nsp* 14 and *N* genes | Hong Kong University | *Anonymous (2020c)* |
| Pharyngeal swab | Real-Time RTPCR | *N* gene | National Institute of Infectious Diseases in Japan | *Shirato et al. (2019)* |
| Serum | CRISPR-Cas (RPA) | Nucleic acid biomarker | China | *Wang et al. (2020a)* |
| Nasopharyngeal swabs | CRISPR-Cas (RTRPA) | Nucleic acid biomarker | USA | *Kellner et al. (2019)* |
| Synthetic COVID19 virus RNA fragment | CRISPR-based SHERLOCK (dipstick) | *ORF1ab* and *S* genes | MIT, USA | *Zhang, Abudayyeh & Gootenberg (2020c)* |
| Throat, nasal, nasopharyngeal or oropharyngeal swabs | ID NOW™ COVID-19 | *RdRp* gene | Abbott | https://bit.ly/3b0W8bd |
| Human finger pricks or venous whole blood, serum, and plasma | Immunoassay | IgM/IgG | BioMedomics, USA | https://bit.ly/2UXh5OF |
| Human finger pricks or venous whole blood, serum, and plasma | Immunoassay | IgM/IgG | China | *Lin et al. (2020)* |
| Human finger pricks or venous whole blood, serum, and plasma | Immunoassay | IgM/IgG | Diazyme | https://bit.ly/2UXlils |

**Notes.**

[a]Nasopharyngeal or oropharyngeal swabs, sputum, lower respiratory tract aspirates, 11 bronchoalveolar lavage, and nasopharyngeal wash/aspirate or nasal aspirate; RPA, 12 recombinase polymerase amplification.

antigens are not yet widely used during this pandemic probably due to longer time (7-14 days) required for the detectable antibodies in the patient's blood. In fact, production of antibody in human bloods requires weeks after infection by the SARS-CoV-2 which limits the use of antibody-based test methods for the early detection of the disease. A research group of Peking University developed a new method for rapid construction of transcriptome sequencing library of Sequencing HEteRo RNA-DNA-hYbrid (SHERRY), which is helpful for rapid sequencing of SARS-CoV-2 (*Di et al., 2020*). They showed that Tn5 transposase, which randomly binds and cuts double-stranded DNA, can directly fragment and prime the RNA/DNA heteroduplexes generated by reverse transcription. The primed fragments are then subject to PCR amplification. This provides an approach for simple and accurate RNA characterization and quantification.

The recent outbreak of the SARS-CoV-2 can be diagnosed using qPCR, but inadequate access to reagents and equipment has slowed disease detection. To rapidly diagnose the disease, Zhang group of MIT developed a test paper for rapid detection of SARS-CoV-2 in one hour by using SHERLOCK (**S**pecific **H**igh Sensitivity **E**nzyme **R**eporter Un**LOCK**ing) technology. This technology may be used widely after clinical trials (*Zhang, Abudayyeh & Gootenberg, 2020*). This technique used synthetic SARS-CoV-2 *S* and *ORF1ab* genes for the diagnosis and no clinical specimen has yet been tested.

In the process of the development of new technique, an exciting improvement is the DZ-Lite SARS-CoV-2 CLIA IgM and IgG tests established by Diazyme, USA. This technique has received FDA EUA approval (https://bit.ly/2UXlils). The molecular principle of this test is a CLIA that run on an automated Diazyme DZ-Lite 3000 Plus chemiluminescence analyzer with a throughput of 50 tests/h. Similarly, Snibe, China, has developed automated CLIA tests on MAGLUMI CLIA analyzers for the detection of IgG and IgM in the patient sample in 30 min (https://bit.ly/2JXGMZm). The major advantages of automated CLIA analyzers based COVID-19 assays compared to rapid LFIA tests is the very high throughput of samples that can be analyzed and the ability to perform more clinical tests for other biomarkers, such as C-reactive protein (CRP), which also need to be monitored in COVID-19 suspects. The rapid, convenient, low cost and specific serological and automated tests are urgently needed to be distributed worldwide especially in the developing countries for testing higher number of patients to tackle this highly contagious disease.

## ANTIVIRALS FOR THE PANDEMIC SARS-COV-2 VIRUS: VACCINES AND THERAPEUTICS

Despite several public health measures such as case isolation, identification and follow-up of contacts, environmental disinfection, social distance, and the use of personal protective equipment have been introduced (*Wei & Ren, 2020*), in the absence of any antivirals (*Kalita et al., 2020*; *Rahman et al., 2020*; *Wang et al., 2020b*), the disease is spreading at an alarming rate. The new cases of active acute infections are being added to the open COVID-19 database such as NCBI, GSAID, and also to the China National Center for Bioinformation 2019 Novel Coronavirus Resource (2019nCoVR) (Fig. 3), every day, as the case count globally skyrockets. Researchers from across the globe are desperately working

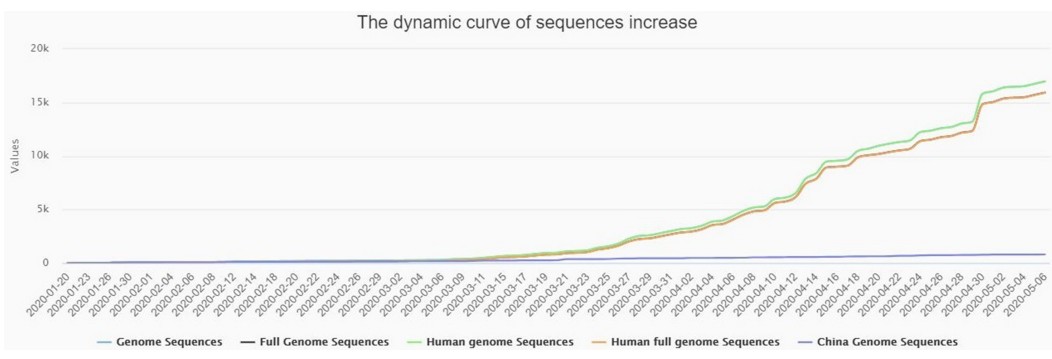

**Figure 3** **The dynamic curve showing daily increase in complete genome sequences of SARS-CoV-2 strain (s) from different patients across the globe, and being submitted to the reference databases.** The data were collected from China National Center for Bioinformation 2019 Novel Coronavirus Resource (2019nCoVR) with available sequences from different countries (as on May 7, 2020).

round the clock to find ways to slow the spread of the novel coronavirus and to find an effective treatment to control this fatal viral disease. Though, more than 200 clinical trials of SARS-CoV-2 treatments or vaccines that are either ongoing or recruiting patients (*Zhou et al., 2020b*), till now no recommended therapeutic drug or vaccines are available for the treatment of COVID–19. The WHO suggested and acknowledged the enormous possibilities of drug repurposing approach. As for example, in the mid of March 2020, the WHO announced the 'SOLIDARITY' clinical trial for COVID-19 treatments (*Khan et al., 2020b*).

At the outset of the epidemic in Wuhan, China, COVID-19 confirmed patients were treated with interferons-$\alpha$ nebulization, broad-spectrum antibiotics, and few antiviral drugs to reduce the viral load (*Shereen et al., 2020*; *Wang et al., 2020b*), however, only remdesivir (GS-5734) has shown promising impact against the virus (*Wang et al., 2020b*). Since then, various other antiviral drugs including nafamostat, nitazoxanide, ritonavir, aak1, baricitinib, arbidol, ribavirin, penciclovir, chloroquine, favipiravir (T-750) or avigan, hydroxychloroquine and chloroquine EIDD-2801 are being tested in clinical trials (*Martinez, 2020*; *Liu et al., 2020*; *Wang et al., 2020b*). The Food and Drug Administration (FDA) announced the cancellation of the use of hydroxychloroquine in the emergency treatment of coronavirus since this anti-malarial drug can cause serious side effects in patients with having health risks. Clinical trials with the nucleotide analog remdesivir (ClinicalTrials.gov: NCT04257656, NCT04252664, NCT04280705), and protease inhibitors (ClinicalTrials.gov: NCT04255017, NCT04276688) have been done in China and the United States. Remdesivir works against coronaviruses closely related to SARS-CoV-2 in animal models (*de Wit et al., 2020*; *Sheahan et al., 2020a*). Remdesivir's mechanism of action as a nucleotide analog is not clear, however, it targets viral RNA polymerase, and terminates RNA synthesis, leads to incorporation mutagenesis, or both (*Amanat & Krammer, 2020*). In addition, a combination of the two protease inhibitors, lopinavir and ritonavir, are also being tested in clinical trials (e.g., ClinicalTrials.gov: NCT04264858), and these drugs can inhibit the cytochrome P450 (*Amanat & Krammer, 2020*; *Cao et al.,*

*2020*). Antiviral arbidol, a fusion inhibitor has also been under ongoing clinical trials (ClinicalTrials.gov: NCT04287686), and dosing with this drug may act through human ACE2 receptor to neutralize the virus, and prevent lung damage (*Amanat & Krammer, 2020*). Another interesting option is the use of convalescent serum as treatment; clinical trials to test this are ongoing in China (ClinicalTrials.gov: NCT04264858, placebo control, not recruiting yet), and compassionate use of this strategy has recently started in the US (e.g., at Mount Sinai Medical Center, NY). Likewise, transgenic cows derived polyclonal human immunoglobulin G (IgG) could be used, and has been tested for safety in clinical trials (ClinicalTrials.gov: NCT02788188). This strategy was successful for MERS-CoV in animal models (*Luke et al., 2016*). Many of these trials will have results within few months, and if remdesivir (produced by Gilead) and/or lopinavir plus ritonavir (produced by AbbVie as Kaletra and Aluvia, respectively) show effectiveness, they could potentially be used widely. Considerate use of these drugs has already been reported for SARS-CoV-2 infections (*Holshue et al., 2020*). The orally bioavailable modified nucleoside analog, $\beta$-D-N4-hydroxycytidine (NHC, EIDD-1931), is a broad-spectrum antiviral drug against various unrelated RNA viruses including influenza, Ebola, CoV, and Venezuelan equine encephalitis virus (VEEV) (*Reynard et al., 2015*; *Agostini et al., 2019*; *Toots et al., 2019*). This proven NHC/EIDD-2801 against multiple coronaviruses showed potential antiviral activity against SARS-CoV-2, and recommended for future zoonotic outbreaks of coronaviruses (*Sheahan et al., 2020b*). Dexamethasone being a steroid reduces inflammation and suppressing immune activation of immune agents, could be inducing the anti-inflammatory effects, and reducing the secretion of cytokines into the lungs (*Kupferschmidt, 2020*). In a recent recovery trial, COVID-19 patients who received dexamethasone for 10 days had reduced deaths by one-third (*Kupferschmidt, 2020*). Despite, there are several reports of using corticosteroids in the treatment of SARS-CoV-2, the available data on safety and efficacy of corticosteroids in COVID-19 is controversial since it can delay virus clearing (*Li et al., 2020b*).

Immunoprophylaxis through passive transfer of antibodies is regarded as an effective method for clinical treatment of infectious diseases. For example, the use of versatile class of mAbs is a new era in infectious disease prevention. This passive immunization overcomes many drawbacks associated with serum therapy and intravenous immunoglobulins preparations in terms of specificity, purity, low risk of blood-borne pathogen contamination and safety (*Ter Meulen et al., 2006*; *Shanmugaraj et al., 2020*). Several earlier studies reported the successful generation of neutralizing antibodies in mice against SARS-CoV through experimental vaccination or passive transfer of mAb, and subsequent reduction of viral replication (*Traggiai et al., 2004*; *Sui et al., 2005*; *Ter Meulen et al., 2006*). Thus, mAbs with potent neutralizing activity against SARS-CoV-2 infections could become promising candidates for both prophylactic and therapeutic interventions (*Shanmugaraj et al., 2020*; *Zhou et al., 2020b*). Though several polyclonal antibodies from recovered SARS-CoV-2-infected patients have been used to treat SARS-CoV-2 infection, but no SARS-CoV-2-specific neutralizing monoclonal antibodies (mAbs) have been reported so far. Researches are ongoing to develop mAbs and/or their functional fragments as putative prophylactic or therapeutic agents to prevent SARS-CoV-2 infections (*Jiang, Hillyer & Du,*

*2020*). The genome of the SARS-CoV-2 virus is closely related to SARS-CoV, and their spike proteins share more than 75% aa sequence identity (*Rahman et al. , 2020*; *Yuan et al., 2020*; *Zhou et al., 2020b*). Researchers have attempted to discover SARS-CoV natural antibodies (nAbs) with potential cross-reactivity, and/or cross-neutralizing activity against SARS-CoV-2 infections (*Jiang, Hillyer & Du, 2020*). Remarkably, a SARS-CoV-specific human mAb, CR3022, could bind potently with 2019-nCoV RBD (KD of 6.3 nM), and recognize an epitope on the RBD that does not overlap with the ACE2-binding site (*Tian et al., 2020*). Although, some of the potent SARS-CoV-specific neutralizing antibodies (e.g., m396, CR3014) that target the ACE2 binding site failed to bind SARS-CoV-2 S protein, the CR3022 might have the potential to be developed as candidate therapeutics, alone or in combination with other nAbs, for the prevention and treatment of SARS-CoV-2 infections (*Tian et al., 2020*). Furthermore, SARS-CoV RBD-specific polyclonal antibodies have cross-reacted with the SARS-CoV-2 RBD protein, and cross-neutralized SARS-CoV-2 infection in HEK293T cell line firmly expressing the human ACE2 receptor, opening avenues for the development of SARS-CoV RBD-based vaccines that might eventually prevent SARS-CoV-2 and SARS-CoV infection (*Jiang, Hillyer & Du, 2020*). A human mAb, 47D11, has been developed that binds to a conserved epitope on the spike RBD, and has the ability to cross-neutralize SARS-CoV and SARS-CoV-2 through a mechanism of receptor-binding inhibition (*Wang et al., 2020d*). This antibody (47D11) would be useful for development of antigen detection tests, and serological assays targeting SARS-CoV-2 (*Wang et al., 2020d*). It is plausible that SARS-CoV RBD-targeting nAbs could be applied for prophylaxis and treatment of SARS-CoV-2 infection in the absence of SARS-CoV-2-specific vaccines and antibodies, but demands for robust testing. Even as the hunt for a vaccine to treat COVID-19 continues, a classic adaptive immunotherapy known as convalescent plasma (CP) therapy that was successfully applied over the past two decades to treat SARS, MERS, and 2009 H1N1 outbreaks with satisfactory efficacy and safety (*Cheng et al., 2005*; *Hung et al., 2011*; *Ko et al., 2018*) holds good promise. In a recent pilot study, Duan et al. reported that CP therapy was found to be well tolerated and could potentially improve the clinical outcomes through neutralizing viremia in severe COVID-19 cases (*Duan et al., 2020*). One dose of CP with a high concentration of neutralizing antibodies can rapidly reduce the viral load, and tends to improve clinical outcomes. However, the optimal dose and treatment time point, as well as the definite clinical benefits of CP therapy should be further investigated in randomized clinical studies.

Vaccines are the most effective and economical means to prevent and control the infectious viral diseases (*Zhang et al., 2020a*). There are multiple attempts in progress to develop such a vaccine following previously described strategies for SARS-CoV and MERS-CoV which might be effective against SARS-CoV-2. Currently, more than 90 vaccines are being developed against SARS-CoV-2 by different research teams in companies and universities across the world. Major vaccine platforms include traditional recombinant protein, replicating and non-replicating viral vectors, and nucleic acid DNA and mRNA approaches (*Corey et al., 2020*). At least six groups have already begun injecting formulations into volunteers in safety trials; others have started testing in animals. A research group led by Professor Sarah Gilbert of Oxford University developed an adenovirus
(ChAdOx1)-based vaccine, the 'ChAdOx1 nCoV-19' targeting the spike protein of the SARS-CoV-2, and two healthy volunteers have been immunized on 24 April, 2020 as the first clinical trial of this vaccine (*Lane, 2020*). *Gao et al. (2020)* have developed an inactivated vaccine candidate (PiCoVacc), which induced SARS-CoV-2-specific neutralizing antibodies in mice, rats and non-human primates. However, inactivated and attenuated virus vaccines have a wide range of disadvantages and side effects including inappropriate for highly immunosuppressed individuals (*Shang et al., 2020b*), phenotypic or genotypic reversion is possible and can still cause some disease (*Regla-Nava et al., 2015*). Alternatively, putative protective antigen/peptides vaccine candidate for SARS-CoV-2 should be considered on the basis immunogenicity (*Wang et al., 2020d*). Moreover, subunit vaccines may be target specific, well-defined neutralizing epitopes with improved immunogenicity and/or efficacy (*Zhang et al., 2020a*; *Wang et al., 2020e*).

With the advancement in immunoinformatics and computational biology, it is now possible to accelerate the vaccine development (*Rahman et al., 2020*; *Zhang et al., 2020a*), and these methods have surpassed the conventional methods. Quite a good number of vaccines are in the pipeline against SARS-CoV-2. An mRNA-based vaccine (mRNA-1273) co-developed by Moderna (a company based in Cambridge, Massachusetts) and the Vaccine Research Center at the National Institutes of Health, like many of the other SARS-CoV-2 vaccines in development, is designed to train the immune system to make antibodies that recognize and block the S protein that the virus uses to enter human cells (*Callaway, Cyranoski & Mallapaty, 2020*). And this vaccine is currently the furthest along, and has already started the phase I trial (ClinicalTrials.gov: NCT04283461) in human and animals (*Amanat & Krammer, 2020*). Preclinical trials of another DNA-based vaccine candidate, INO-4800 demonstrated as a promising candidate to protect against the novel coronavirus SARS-CoV-2 (*Inovio, 2020*). INO-4800 targets the major surface antigen S protein of SARS-CoV-2 virus, and induced antibodies to block SARS-CoV-2 S binding to the host ACE2 receptor. Vaccination with INO-4800 generated near-100% seroconversion, robust binding and neutralizing antibody as well as T cell responses in mice and guinea pigs (*Inovio, 2020*). The Centre for Disease Control and Prevention (CDC), China is working to develop an inactivated virus vaccine (*Cheung et al., 2020*). An mRNA-based vaccine's sample prepared by Stermirna Therapeutics will be available soon (*Xinhua, 2020*). The GeoVax and BravoVax (Wuhan, China) is working to develop a Modified Vaccina Ankara (MVA) based vaccine (*Geo-Vax, 2020*). In addition, the Clover Biopharmaceuticals is trying to develop a recombinant 2019-nCoV S protein subunit-trimer based vaccine (*Clover, 2020*). Another biopharmaceutical company Curevac (Germany) is working on a similar vaccine but is still in the pre-clinical phase. Additional approaches in the pre-clinical stage include viral-vector-based vaccines (focused on the S protein, e.g., Vaxart, Geovax, University of Oxford, and Cansino Biologics), recombinant-protein-based vaccines (focused on the S protein, e.g., ExpresS2ion, iBio, Novavax, Baylor College of Medicine, University of Queensland, and Sichuan Clover Biopharmaceuticals), DNA vaccines (focused on the S protein, e.g., Inovio and Applied DNA Sciences), live attenuated vaccines (Codagenix with the Serum Institute of India, etc.), and inactivated virus vaccines (*Amanat & Krammer, 2020*). All of these approaches have advantages and disadvantages, thus, it is not possible
to predict which strategy will be faster or more successful. Two multinational company, Johnson & Johnson (J&J) *Johnson & Johnson (2020)* and *Sanofi (2020)* recently joined efforts to develop SARS-CoV-2 vaccines using an experimental adenovirus vector platform, and a process similar to the process used for their approved Flublok recombinant influenza virus vaccine (*Zhou et al., 2006*). This vaccine may be available within months, if not years, from being ready for use in the human population (*Amanat & Krammer, 2020*).

Being an RNA virus, genome-wide nucleotide mutations and aa mutations and/or substitutions (Table 1) have already been reported in different SARS-CoV-2 strains from across the globe (*Huang et al., 2020*; *Islam et al., 2020*; *Phan, 2020*; *Yin, 2020*; *Wang et al., 2020e*). Therefore, it is critical to develop vaccines with strong efficacy and safety targeting this SARS-CoV-2 to prevent its infection in humans. The structural divergence in the RBD and NTD segments of the S protein in SARS-CoV-2 is main focus of vaccine candidate designing, selection, and development (*Rahman et al., 2020*). Therefore, multi-epitope based vaccines targeting the full-length S protein and its structural domains (RBD, NTD, S1 and S2 subunits), M, E and N proteins can play a great role in fighting against this SARS-Cov-2 virus rather than a single-epitope vaccine (*Rahman et al., 2020*; *Zhang et al., 2020a*).

## CONCLUSIONS AND PERSPECTIVES

The emergence of the novel, pathogenic SARS-CoV-2 in the Wuhan city of China in December 2019, and its rapid national and international spread has created a global health emergency. Genome sequences of a large number of strains of SARS-CoV-2 have been published, and all the research data on this new virus are publicly available. The genomic features described in this review is based on the recent reports of the infectiousness and transmissibility of SARS-CoV-2 in humans. Currently, evidence supports the natural zoonotic origin of the SARS-CoV-2, not a purposefully manipulated laboratory product. Moreover, identifying the closest viral relatives of SARS-CoV-2 circulating in animals will greatly assist future studies of viral function. Indeed, the availability of the RaTG13 bat and Malayan Pangolin sequences helped reveal key RBD mutations and the polybasic furin cleavage sites. Genome-wide annotations of a wider range of sequences (50–2500) revealed considerable number of mutations throughout the SARS-CoV-2 genome, which includes both mismatch and deletion mutations both in translated and untranslated regions. Moreover, the identification of the conformational changes in mutated protein structures and untranslated cis-acting elements is of significance for studying the virulence, pathogenicity and transmissibility of SARS-CoV-2. The discovery of specific diagnostic tool targeting specific genes of the genome of SARS-CoV-2 within weeks of the outbreak of the disease in China was a phenomenal research success which has been playing vital role in tackling this highly contagious disease. Although real-time RT-PCR methods targeting specific genes have widely been used to diagnose the SARS-CoV-2 infected patients, however, recently developed more convenient, rapid, and specific diagnostic tools targeting IgM/IgG or newly developed plug and play methods should be available especially for the resource-poor developing countries. Therefore, the development of

an effective vaccine is one of the most pressing needs to contain the ongoing pandemic of SARS-CoV-2, to reducing morbidity and mortality in infected population, and also preparation for long term prevalence of the SARS-CoV-2 virus. Several approaches for vaccines and antivirals targeting human coronaviruses are in developmental stages, which could be safely and effectively used against the current as well as future epidemics. We can assume that potential targets for development of drugs and multiepitope-based chimeric peptide vaccines against this newly emerging lineage B beta-CoV, SARS-CoV-2 will be available within a reasonable period of time. However, vaccine delivery modality and immunization strategy should be ensured through rapid human and animal-based trails before commercialization. Nevertheless, owing to the different experimental methods, sample sizes, sample sources, and research perspectives of various studies, results have been inconsistent, or relate to an isolated aspect of the virus or the disease it causes. At present, systematic summary data on the SARS-CoV-2 are limited. This review summarizes new knowledge on genomics, genome evolution, developed diagnostic methods and progress in development of vaccine or therapeutics, from multiple perspectives, with the aim of gaining a better overall understanding, prevention and control of the disease. This review also discusses on scopes for further research and effective management and surveillance of the COVID-19 pandemics.

### Funding
The authors received no funding for this review article.

### Competing Interests
The authors declare that they have no financial, non-financial, professional or personal competing interests.

### Author Contributions
- M. Nazmul Hoque conceived and designed the experiments, performed the experiments, analyzed the data, prepared figures and/or tables, authored or reviewed drafts of the paper, and approved the final draft.
- Abed Chaudhury, Md Abdul Mannan Akanda and M. Anwar Hossain performed the experiments, analyzed the data, authored or reviewed drafts of the paper, and approved the final draft.
- Md Tofazzal Islam conceived and designed the experiments, performed the experiments, analyzed the data, authored or reviewed drafts of the paper, and approved the final draft.

### Data Availability
This is a Literature Review article without any data or code.

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
