# Peer review of "Genomic diversity and evolution, diagnosis, prevention, and therapeutics of the pandemic COVID-19 disease"

_PeerJ, doi:10.7717/peerj.9689_

## Round 0.1 · original submission · Major Revisions

Please, answer all comments, questions and suggestions of the reviewers. Pay a special attention to the proposal made by Reviewer 1:
"... at this point the scope of this manuscript is too broad for a journal paper. I would recommend the authors should concentrate on mutation/evolution aspect of this paper because it is developed best.

·

Basic reporting

This manuscript presents an attempt of comprehensive representation of the literature concerning SARS-CoV-2, with three-pronged approach: 1) structure and evolution, with a review of mutational landscape; 2) treatments; 3) development of the vaccines. While this manuscript is having certain educational value in its broad coverage, at this point its scope is too broad for a journal paper. I would recommend selecting just one of these "topical areas" and go further in depths. If I may suggest, mutation/evolution aspect of this paper is developed best, may be authors should concentrate on that, while making forays into adjacent areas (i.e. connection between mutations in particular epitopes and vaccine properties of these epitopes). Also, diagnostics approaches may be included, as they are directly related to analysis of viral sequences.

As it is written now, the treatments portion is very sketchy and does not go beyond naming candidate treatments, and the vaccine portions somehow not discussing natural anticoronavirus immunity, and recent studies about antibody levels and strengths of immunity post-infection. These parts are underdeveloped, and, therefore, should be reserved to future. separate, manuscripts.

Experimental design

the Review is well organized, but is too broad. In its broadness, it is missing important aspects (i.e. immunity, mechanisms of antiviral action for named compounds). I propose that authors should narrow the scope while increasing the depths and interpretation

Validity of the findings

as such, this study does not argue any particular point of view except stating that coronavirus research is actively developing area. Because of that, conclusions and perspectives part seem trivial. Also, Fig.1 is completely unneccessary.

Additional comments

this is a good effort, which needs a bit of focusing before publication. One cannot cover entire Universe, one will be spread too thin!

·

Basic reporting

no comment

Experimental design

no comment

Validity of the findings

no comment

Additional comments

The review (article Id 48746) submitted by Houque et al. reported on i) Genome analysis and evolution of SARS-Cov-2, ii) diagnostic tools and iii) updating the therapeutics and vaccine development against COVID19. The review is well organized and well written and I recommend it publication in PeerJ. I have two main comments:
1- The authors mentioned that the S-Protein of SARS-Cov-2 possesses furin cleavage site at S1 and S2 boundary (Walls et al., 2020). This finding was also published by Coutard et al (2020). This furin-like cleavage is absent in betacoronaviruses of lineage b including SARS-CoV sequences. I suggest if the author can discuss this findings deeper to shed lights on the potential origin (Can this a natural evolution for Coronaviruses?), the consequences, and the impacts on treatment. Based on this finding, is SARS-CoV-2 a manipulated laboratory construct?
I do not believe also that any type of laboratory-based scenario is plausible, based on the available data, but I think it will be great if the authors discussed this point in more details.
These links might be helpful (of course only publications can be cited)
Andersen, K.G., Rambaut, A., Lipkin, W.I. et al. (2020): The proximal origin of SARS-CoV-2. Nat Med 26, 450–452.
Coutard et al. (2020): The spike glycoprotein of the new coronavirus 2019-nCoV contains a furin-like cleavage site absent in CoV of the same clade. Antiviral Research 176: 104742
Follis, K. E., York, J. & Nunberg, J. H. Virology 350, 358–369 (2006)
https://nerdhaspower.weebly.com/blog/scientific-evidence-and-logic-behind-the-claim-that-the-wuhan-coronavirus-is-man-made
https://medium.com/@yurideigin/lab-made-cov2-genealogy-through-the-lens-of-gain-of-function-research-f96dd7413748

2- Figure (1), I suggest to delete it. The review is very rich and there is no need for this figure.

Other comments:
- Line 83: please add the reference
- Line 181: Please delete the word spike (use the abbreviation “S”), the full name was mentioned in line 103
- Line 130: Please delete the word “and” before antifungal treatment
- Line 143: Again S protein Not spike protein
- Line 173: Please use the abbreviation ICTV, the full name was mentioned in line 63
- Lines 187-188: Please use abbreviations of S, N, E, and M; also please delete fig 1.
- Line 191: Please delete “open reading frame” and use abbreviation, it is mentioned in line 85
- Line 196: 29,844 (coma)
- Line 197: 29,752 and 30, 119 (coma)
- Line 197 (add aa after 7,078 to be 7,078 aa)
- Line 201-202: use abbreviations E, M, and N
- Line 202: please use a uniform writing: beta coronavirus, betacoronavirus or beta-coronavirus?
- Line 237: S
- Line 241: coma after the word respectively
- Line 253: should be: National Center for Biotechnology Information (NCBI)
- Lines 258-260: Please explain your findings in more details
- Line 272: coma after respectively.
-
-

·

Basic reporting

Please refer to general comments to authors. Issues noted.

Experimental design

Please refer to general comments to authors. Issues noted.

Validity of the findings

Please refer to general comments to authors. Issues noted.

Additional comments

In this manuscript, the authors’ goal is to write a “comprehensive review on genomic diversity and evolution, diagnosis, prevention, and therapeutics of the SARS-CoV-2.” The overall constructive criticism is that it is an overambitious review with frequent superfluous sentences. There are many major points for authors’ consideration.

• The review contains many superfluous sentences that do not add critical analysis or provide scientific insights. Example in Lines 605- , “We expect researchers and/or companies who are working round the clock will bring a new SARS-CoV-2-based vaccine from isolated virus particle or gene sequences to clinical testing within a short period of time.”
• COVID-19 is the disease, and SARS-CoV-2 is the virus. The terms are not interchangeable as authors suggested.
• Lines 184-185: “The positively-sensed single-stranded RNA SARS-CoV-2 virus (Ahmed et al., 2020) has a genome size of 26,000 and 32,000 bases (Abdelmageed et al., 2020). It shares more than 80% sequence identity to the previously reported human coronaviruses.” The genome size that the authors provide is the range for the coronavirus genus. Authors should look carefully to precise data on the SARS-CoV-2 genome length.
• The authors stated: “Genomic analyses have indicated that the virus, popularly named as corona, is probably not a purposefully manipulated laboratory construct.” “Corona” may be is considered a popular name for the virus these days, as authors claimed. However, this information should not be considered in a scientific review. In addition, the authors lost the opportunity to discuss the scientific reasons why SARS-CoV-2 is unlikely a laboratory construct.
• There are many paragraphs that lack focus. As an example, lines 113-119. Do the authors want to write on entry, virulence, or epidemiology?
• Figure 2 needs some revision. ORF is not present in all human pathogenic CoVs. Also, in MERS, ORF4a is shown twice; please correct the second for 4b.
• Figure 4 shows “The dynamic curve showing daily increase in complete genome sequences of SARS-CoV-2 strain (s) from different patients across the globe, and being submitted to the reference databases. The data were collected from China National Center for Bioinformation 2019.” However, in Lines 462-464, authors mentioned Fig 4 as “The new cases of active acute infections are being added to the open COVID-19 database every day.
• The manuscript must be edited throughout. There are multiple grammatical errors, including mistakes in the use of “are” vs. “is” are very frequent. Example: Lines 115-116: “However, the virulence mechanisms of the SARS-CoV-2 is not yet clear.”

---

## Round 0.2 · accepted · Accept

I agree with the reviewers that it would be possible for further improvement of the manuscript; however, I can summarize that the manuscript is informative enough as a literature review. Taking into account the importance of the topic I would recommend it's publication.

·

Basic reporting

no comment

Experimental design

no comment

Validity of the findings

no comment

Additional comments

no comment